# A high-current hydrogel generator with engineered mechanoionic asymmetry

Hongzhen Liu [1], Xianglin Ji[2,3], Zihao Guo[4], Xi Wei[1], Jinchen Fan[5], Peng Shi [2,3], Xiong Pu [4] ✉, Feng Gong [6] ✉ & Lizhi Xu [1,7] ✉

Mechanoelectrical energy conversion is a potential solution for the power supply of miniaturized wearable and implantable systems; yet it remains challenging due to limited current output when exploiting low-frequency motions with soft devices. We report a design of a hydrogel generator with mechanoionic current generation amplified by orders of magnitudes with engineered structural and chemical asymmetry. Under compressive loading, relief structures in the hydrogel intensify net ion fluxes induced by deformation gradient, which synergize with asymmetric ion adsorption characteristics of the electrodes and distinct diffusivity of cations and anions in the hydrogel matrix. This engineered mechanoionic process can yield 4 mA (5.5 A m$^{-2}$) of peak current under cyclic compression of 80 kPa applied at 0.1 Hz, with the transferred charge reaching up to 916 mC m$^{-2}$ per cycle. The high current output of this miniaturized hydrogel generator is beneficial for the powering of wearable devices, as exemplified by a controlled drug-releasing system for wound healing. The demonstrated mechanisms for amplifying mechanoionic effect will enable further designs for a variety of self-powered biomedical systems.

Progress towards the intimate integration of electronics into organisms calls for soft and biocompatible mechanoelectrical energy converters, which could either be responsive to mechanical stimuli for physiological sensing/monitoring or be able to harvest the rich biomechanical energy for power supply[1–3]. Conventional electromagnetic generators, despite the large electrical outputs at relatively high frequencies, obviously fall short due to the large mechanical mismatch with the biological systems[4,5]. The substantially developed triboelectric and piezoelectric nanogenerators, capable of employing various soft materials, are, nevertheless, characterized by high internal impedance and low output current[6–8]. Given the facts that most biomechanical energy originating from human physiological activities are low-frequency and low-speed[9,10], whereas some significant biomedical

applications, like muscle electrical stimulation[11,12] and bone regeneration[13], require high electrical currents, it is of particular importance to develop soft mechanoelectrical devices that can yield high current and high transferred charge at low-frequency and low-speed mechanical energy inputs.

Current generation through mechanoionic mechanisms is a promising approach to meeting these requirements[14–19]. In analogous to piezoelectric mechanism where the dipoles in dielectric materials are polarized by mechanical pressure, electrolytic materials, such as hydrogels, can also be polarized mechanically due to the deformation-induced unbalanced distribution of cations and anions. The essence is to produce gradient deformation, and therefore, an asymmetric cation/anion distribution at the two-electrode/hydrogel interfaces.

[1]Department of Mechanical Engineering, The University of Hong Kong, Hong Kong SAR, China. [2]Department of Biomedical Engineering, City University of Hong Kong, Hong Kong SAR, China. [3]Hong Kong Centre for Cerebro-Cardiovascular Health Engineering, Hong Kong Science Park, Hong Kong SAR, China. [4]Beijing Institute of Nanoenergy and Nanosystems, Chinese Academy of Sciences, Beijing, China. [5]School of Materials and Chemistry, University of Shanghai for Science and Technology, Shanghai, China. [6]Key Laboratory of Energy Thermal Conversion and Control of Ministry of Education, School of Energy and Environment, Southeast University, Nanjing, China. [7]Advanced Biomedical Instrumentation Centre, Hong Kong Science Park, Shatin, New Territories, Hong Kong SAR, China. ✉e-mail: puxiong@binn.cas.cn; gongfeng@seu.edu.cn; xulizhi@hku.hk

This asymmetry can be induced by a streaming current of polyelectrolyte hydrogels with single-sign free ions[18,19], or produced in neutral hydrogels with cations and anions diffusing at different rates. The latter mechanism was recently demonstrated as a piezoionic effect[20]. Actually, analogous asymmetric ion distribution/diffusion has also been achieved thermally via temperature gradient for thermoelectric generation[21], and chemically via functional groups gradient[22] for moisture-induced[23,24] or water-evaporation-induced electricity generation[25,26]. Nevertheless, the achievable current output from mechanoionic effect remains at a low level of ~1 µA[20], which is not sufficient as a power supply for most practical devices[27].

In this work, we achieved unusually high mechanoionic electricity generation by the synergy of enhanced structural and chemical asymmetry in a hydrogel generator, as depicted in Fig. 1a. A pyramid-structured hydrogel was designed to induce high deformation gradient under compression and, therefore, a net ionic current generation due to the unbalanced diffusion of cations and anions at different rates. This mechanoionic current generation was further amplified by the two electrodes with different surface functional groups and, therefore, distinct properties of ion adsorption at the electrode/hydrogel interfaces. The structural and chemical asymmetry of the hydrogel generator collectively contributes to the enlarged electricity

generation, yielding milliampere scale current at 0.1 Hz cyclic compression (0.25 mA at 2 kPa and 4 mA at 80 kPa), transferring 916 mC m$^{-2}$ charge per cycle, far beyond that of the previous reports (Supplementary Table. 1).

## Results
### Design and structures
Figure 1a illustrates the design and principles of the hydrogel generator achieving enhanced mechanoionic electricity generation. The structural asymmetry was realized by designing the hydrogel into a pyramid-shaped array, which was fabricated by pouring 15 wt% PVA solution into a 3D-printed mold and demolding after physical cross-linking. Subsequently, the crosslinked hydrogel was loaded with mobile ions by immersing it in 4 M LiCl solution for 1 day before use. The section area of the hydrogel pyramid changes with the height, resulting in a high-strain gradient under normal compression, as verified by finite element analysis (FEA) (Fig. 1c). This strain gradient displaces the water from the region with large compressive strain to that with smaller strain, and then the resulting gradient of the ion concentration leads to the cation/anion diffusion along the strain gradient. Due to the different diffusivity of cations and anions, a net ionic current will be generated inside the hydrogel and, therefore, an electric

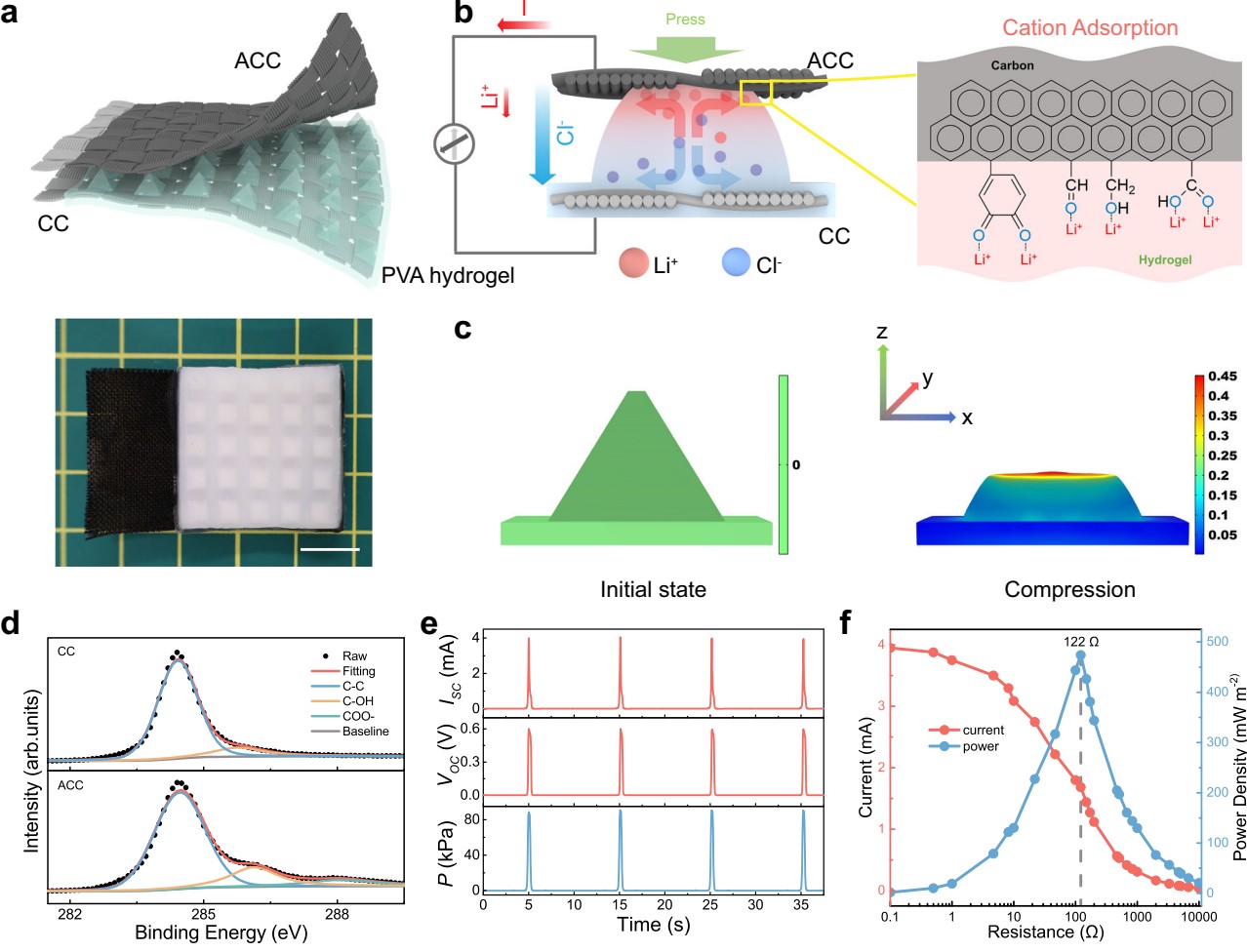

**Fig. 1 | Mechanoionic asymmetry and high current output of the hydrogel generator. a** A schematic illustration of the device (top) and a photograph of the CC-PVA part. The device contains an activated carbon cloth (ACC) as the working electrode, a PVA hydrogel with pyramid arrays, and a raw carbon cloth (CC) as the counter electrode. Scale bar: 1 cm. **b** The mechanism of mechanoionic current generation. The strain gradient displaces Cl$^-$ (blue) with a higher diffusivity than that of Li$^+$ (red). On the other hand, the ACC possesses stronger adsorption to Li$^+$

cations than Cl$^-$ anions, leading to an enhanced separation of charges. The enlarged figure shows the electrode-ion interaction at the ACC/hydrogel interface. **c** FEA-simulated strain before (left) and after (right) applying compression to the hydrogel structure. **d** C1s XPS spectra for CC and ACC, indicating incorporated oxygen on ACC. **e** Profiles of the generated short-circuit current, open-circuit voltage and applied pressure. **f** Power density and output current as functions of loading resistance.

current through the external circuit. We note that the strain gradient-induced by the pyramid structures (Supplementary Fig. 1) is ~9.5 times higher than that in a uniform hydrogel layer without structuring, which should facilitate the generation of ionic current (see details in Supplementary Fig. 2 and Methods). We further amplify this mechanoionic effect by designing two electrodes with different surface chemistries and ion adsorption capabilities. A piece of carbon cloth (CC) was used as-received without treatment as the bottom counter electrode; another activated carbon cloth (ACC) undergoing an electrochemical activation process (see experimental details in Methods) was utilized as the top working electrode[28]. The CC counter electrode was immersed inside the hydrogel precursor so as to ensure a robust interface after cross-linking. The abundant oxygen-containing functional groups on the surface of the ACC working electrode were confirmed by the X-ray photoelectron spectroscopy (XPS) (Fig. 1d and Supplementary Fig. 3), and elemental mapping of scanning electron microscopy (SEM) images (Supplementary Fig. 4). The C$1s$ XPS spectrum of the ACC exhibits a much-intensified peak of C-OH species (285.9 eV) and an additional peak of COO- species (288.1 eV)[29], as compared with that of the CC. The total oxygen content for ACC and CC was determined to be 16.95 at% and 3.16 at%, respectively. This functionalized working electrode (ACC) with abundant oxygen is expected to anchor the cations at the interface due to the improved cation adsorption capability, and therefore intensify the net ionic current generation.

The hydrogel generator outputs a large current pulse under compressive deformation (Fig. 1e). The current can be as high as 4 mA (5.5 A m$^{-2}$) under 80 kPa pressure, with the transferred charge being 916 mC m$^{-2}$, three to five orders of magnitude larger than existing devices (Supplementary Table 1), reaching the threshold for direct electrical stimulation for muscles[30]. Supplementary Fig. 5 and Fig. 1f illustrate the corresponding current, voltage, and power for various loads. The maximum peak power density of 475 mW m$^{-2}$ is achieved with an external load of 122 Ω, suggesting that the internal resistance of the hydrogel generator is less than one thousandth of triboelectric or piezoelectric nanogenerators (TENG or PENG) (Supplementary Table 1). Compared to the recently reported generator based on piezoionic effects[20], this hydrogel generator also shows advantages in the current (4 mA verse 1 μA) and power (475 mW m$^{-2}$ verse 0.85 μW cm$^{-3}$) outputs. The detailed theoretical thermodynamic analysis of the current generation (see details in Supplementary Note) also supports that enlarging the cation and anion diffusion flux is essential for high electrical outputs. We also excluded the contribution of the triboelectric effect to the current output (Supplementary Fig. 6).

## Work mechanism

The asymmetric surface functionalization of the electrodes is playing an important role in the significantly improved mechanoionic electricity generation. Specifically, maximum output is achieved only if differential ion adsorption properties of electrodes are in accordance with the differential ion diffusion in the hydrogel matrix. It is known that Cl⁻ anions diffuse at a higher rate than Li⁺ cations in aqueous solution[31], and this difference in diffusivity is even greater in PVA hydrogels[32]. Therefore, it is highly desired to have a Li⁺-anchoring ACC electrode interfacing with the high-strain region to further reduce the diffusion flux of Li⁺. On the other hand, the Li⁺ anchoring effect is not desired at the counter electrode since it does not help with the separation of cations and anions for electricity generation. Indeed, a set of control experiments well demonstrate these effects. For instance, two generators with symmetric pairs of electrodes (using two CC or ACC electrodes) were tested for comparison, both showing electrical current generation but a much smaller peak value. On the other hand, little generation of current was observed if the ACC working electrode and CC counter electrode were reversed (Fig. 2a), since the ion-anchoring effects were flipped and counteracted with the

differential diffusion of cations and anions. As for the optimized electrode layout, the current and voltage increased with the time duration of electrochemical oxidation of the ACC working electrode (Fig. 2b and Supplementary Fig. 7).

To confirm the role of oxygen-containing functional groups for the anchoring of Li⁺, we calculated the charge redistribution when CC and ACC are in contact with the electrolyte by density functional theory (DFT) (Fig. 2c). The charge redistribution process mainly occurs between H₂O molecules and CC for CC/Li⁺/H₂O system; whereas, for ACC/Li⁺/H₂O system, it is evident that Li⁺ gains electrons while ACC loses electrons. Hence it is not surprising that the adsorption energy of LiCl on ACC (1.2 eV) is higher than that on CC (0.18 eV). For ACC, the adsorption energies of Li⁺ and Cl⁻ are calculated to be 2.515 eV and −0.384 eV (Fig. 2d), respectively. These results indicate that when ACC is in contact with the hydrogel, Li⁺ cations tend to accumulate at the ACC-hydrogel interface, while Cl⁻ anions tend to stay away from the interface. In an experimental investigation, we changed the top working electrode from ACC to gold (Au) film or stainless-steel foil. Oxygen plasma treatment was applied to these metal electrodes for introducing oxygen-containing groups on the surface[33]. As illustrated in Supplementary Figs. 9–S11, it is evident that surface functionalization with abundant oxygen-containing groups of the working electrode can greatly enhance the current output due to the possible anchoring effect for Li⁺. When the PVA hydrogel is replaced with a gelatin methacrylate (GelMA) hydrogel, this phenomenon can still be observed, demonstrating the generality of the proposed mechanism (Supplementary Fig. 12). When using PAAM for the hydrogel layer, however, the output current is significantly lower. In fact, PAAM hydrogel exhibits much greater viscosity than PVA and GelMA hydrogels. The distinct interfacial interactions may affect the surface functionality of ACC, which is worth future investigations[34].

Another set of control experiments further demonstrates the importance of selective ion adsorption on mechanoionic current generation. Specifically, we changed the surface functionalization of the ACC working electrode (Fig. 2e) to be capable of anion adsorption, which is opposite to its optimal design. Polyethyleneimine (PEI), a typical cationic polymer[35], was applied on the surface of the ACC by immersing the ACC into a 25 wt% PEI solution and drying it in an oven. We observed that the direction of current was reversed when changing the working electrode from ACC to ACC-PEI (Fig. 2f). When using ACC as the working electrode and 1 M LiCl hydrogel as the electrolyte in the optimum configuration, a positive peak current of 1.051 mA was obtained at 12 kPa pressure. However, the output changed to an opposite current of −0.111 mA when using ACC-PEI as the working electrode. Subsequently, the ACC-PEI was rinsed with DI water for 2 min to partly remove the PEI and dried again to reduce the PEI content. The response current returned positive with a small value of 0.0349 mA. Further reducing the PEI content by rinsing ACC-PEI for 2 days in DI water recovered the response current back to 0.722 mA. The small amount of PEI residue on the ACC was responsible for the slight decrease in the current compared with the initial value. The variation of PEI content in each step was characterized by FTIR spectroscopy (Fig. 2g and Supplementary Fig. 13). Peaks located at 3250 cm$^{-1}$ for $v$(N-H) and 2925 cm$^{-1}$ and 2830 cm$^{-1}$ for $v$(C-H) were identified as the characteristic peaks of PEI[35]. The changes in the intensity of these peaks indicate the successful modification and the gradual removal of PEI. Furthermore, by employing ACC-PEI as the counter electrode and ACC as the working electrode, we achieved a higher output current compared to that with CC as the counter electrode (Supplementary Fig. 14). These results indicate that the anion adsorption superiority of the ACC-PEI working electrode can overwhelm the strain gradient-induced net anion flux, and lead to the reversed sign of net cation flux. In this regard, the mechanoionic effect can be well tuned by appropriately designing asymmetric chemical functionalization of the electrodes, which is convenient for diverse device applications.

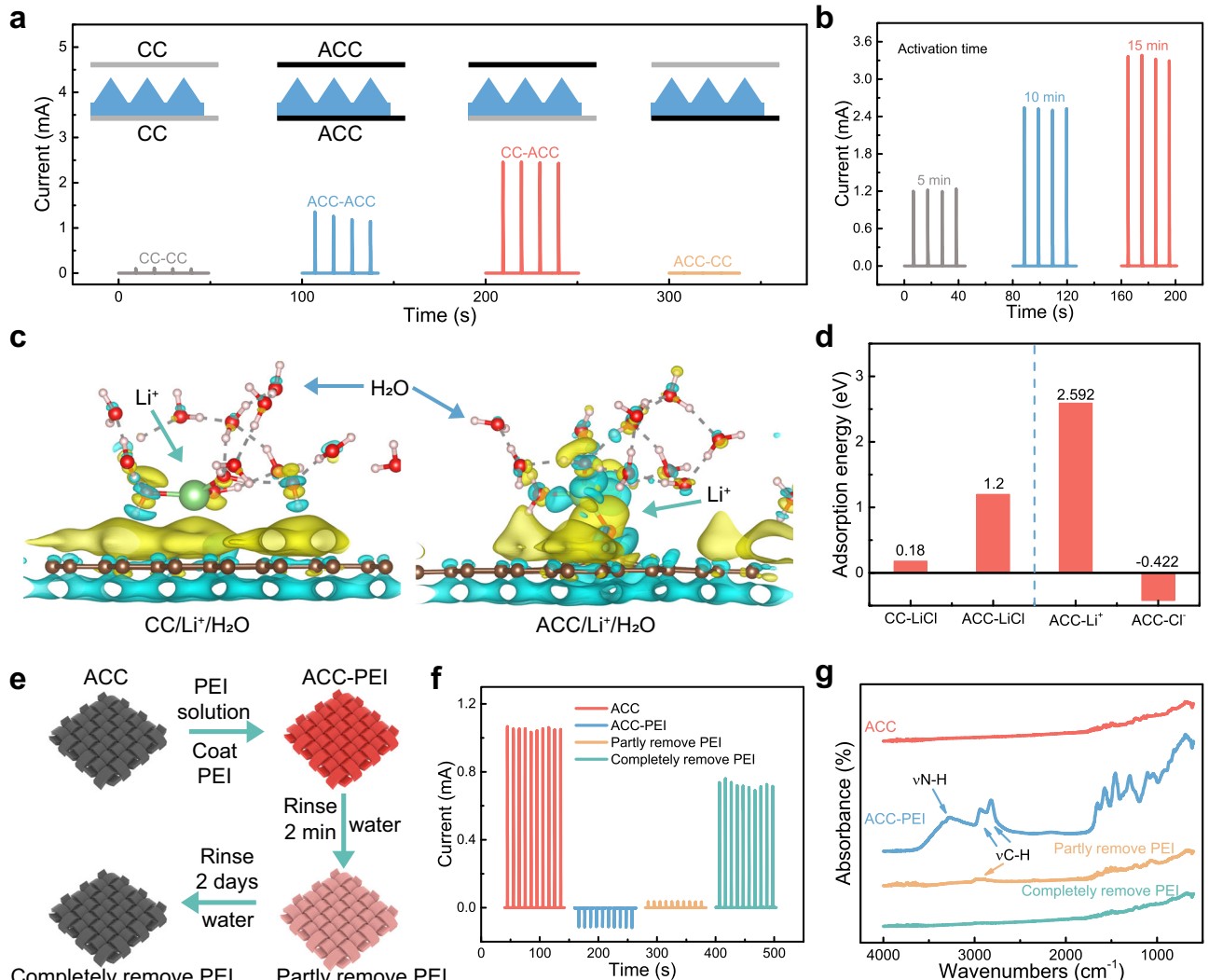

**Fig. 2 | Differential ion adsorption characteristics of the electrodes for enhanced mechanoionic current generation. a** Variation of current outputs when changing the layout of the working and counter electrodes. Devices are named as "counter electrode-working electrode". The highest output current is achieved when ACC serves as the working electrode and CC as the counter electrode. **b** Current outputs of CC-ACC devices with an increased activation time of the ACC electrodes. **c** Simulated charge redistribution in CC/Li+/H₂O system and ACC/Li+/H₂O system (yellow, gaining electron; blue, losing electron). **d** Calculated adsorption energy of the electrodes to various species. **e** Schematic illustration of the process of PEI treatment on ACC, altering the ion adsorption characteristics of ACC. **f** Distinct current outputs of the devices with different treatments of the ACC working electrode. **g** FTIR spectra of ACC at the four stages during the PEI modification process.

## Output performance and applications

With a mechanistic understanding of the devices, we further characterize the performance of these mechanoionic hydrogel generators in the context of practical applications. The optimized electrode layout (ACC as the working electrode and CC as the counter electrode) was used for all the tests thereafter. The influences of ionic concentration and applied pressure on the current output were first studied (Fig. 3a). No current signal was observed under any pressure when DI water-based hydrogel served as the electrolyte. When using hydrogel with 1 M LiCl, the response current went up apparently from 0 mA to 3.3 mA with increasing pressure from 0 kPa to 88 kPa. Higher current output was obtained using hydrogel with 4 M LiCl. The current increased with pressure but reached a saturation of about 4 mA at around 60 kPa.

Next, we studied the influences of recovery time between cycles and strain rates on the current generation. With the loading rate fixed at 16 mm s$^{-1}$ (strain rate: 2 s$^{-1}$), the response current increases with the extension of the recovery time from 1 s to 8 s (frequency:1 Hz to

0.125 Hz) and keeps constant when recovery time exceeds 8 s (Fig. 3b and Supplementary Fig. 17). This phenomenon can be explained as the ions cannot restore the initial uniform distribution within a short recovery time. Transferred charge per pressing-releasing cycle presents the same trend. Even if the recovery time was shortened to 1 s, the amount of transferred charge was up to 180 µC per cycle. In the next experiment, with recovery time fixed at 10.4 s, the strain rate was changed from 2 mm s$^{-1}$ to 30 mm s$^{-1}$, with the corresponding peak current increasing from 0.4 mA to 2.9 mA (Fig. 3c and Supplementary Fig. 18). A higher strain rate leads to a higher gradient in ion concentration due to limited response time for ion redistribution, hence enhancing the differential ionic diffusion flux[36]. On the other hand, a higher strain rate shortens the duration of the current output, reducing the transferred charge per cycle. At a 2 mm s$^{-1}$ strain rate, the transferred charge reached 668 µC (916 mC m$^{-2}$) per cycle, which outperforms the reported mechanoelectrical energy converters[37-40]. Due to the high capability in charge transfer, the hydrogel generator can charge capacitors quickly.

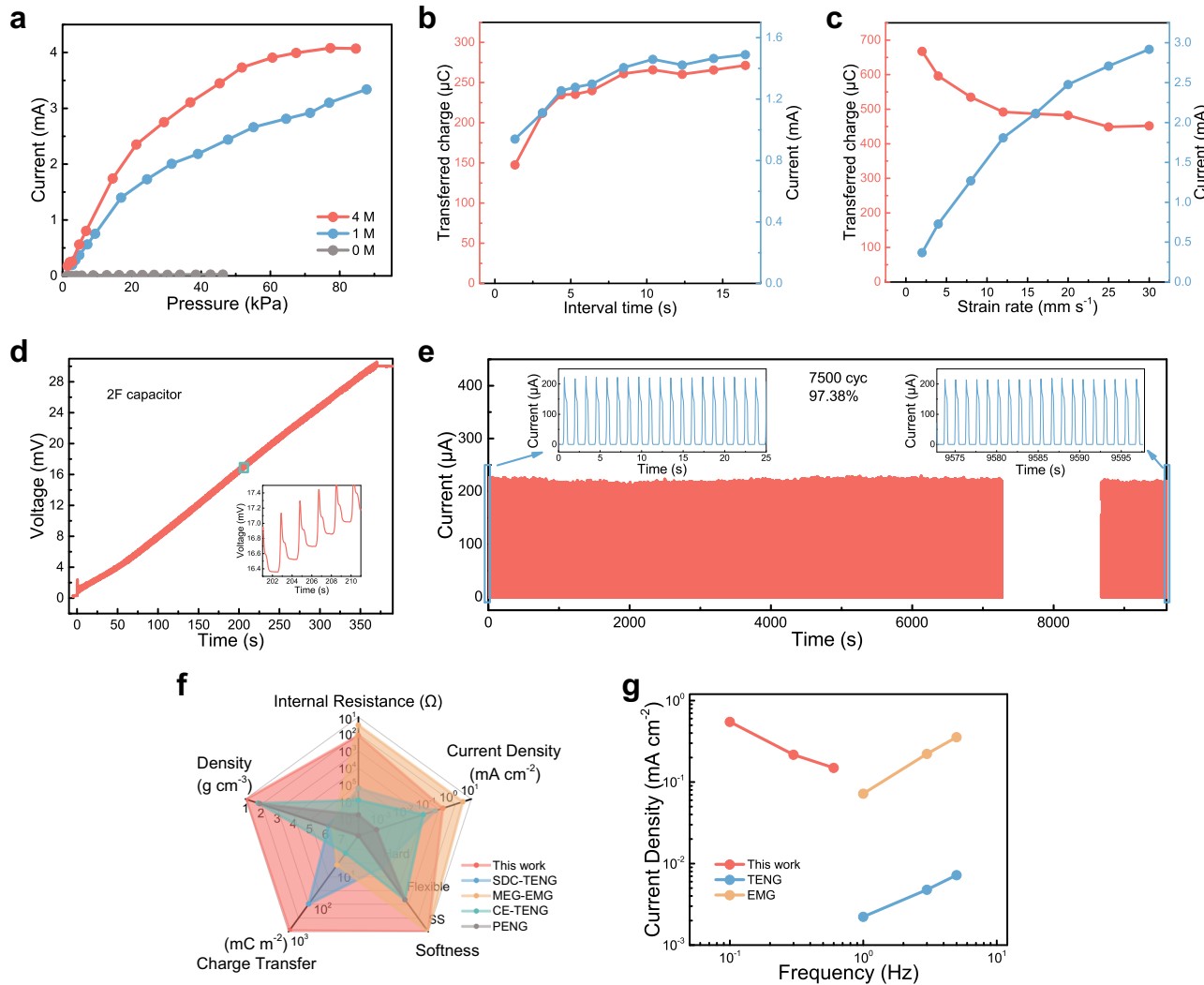

**Fig. 3 | Output performance of the hydrogel generator. a** Peak short-circuit current as a function of applied pressure, with the hydrogels soaked with different electrolytes: DI water (gray), 1 M LiCl (blue) and 4 M LiCl (red). **b** Peak current and charge transferred per cycle as functions of the recovery time between cycles. **c** Peak current and charge transferred per cycle as functions of the strain rate. **d** The charging profile of a 2 F capacitor charged by the hydrogel generator. **e** Current output over 7500 cycles of 5 kPa compression, showing durability and stability of the device. The test parameters are listed in Supplementary Table 3. **f** Comparison of this device with state-of-the-art SDC-TENG[38], EMG[39], TENG[40], and PENG[41] in the domains of internal resistance, current density, softness, transferred charge, and density. **g** Current outputs of this hydrogel generator under various frequencies of mechanical input, as compared with those of TENG and EMG. The test parameters are listed in Supplementary Table 3.

As shown in Fig. 3d, a 2 F capacitor can be charged to 29 mV in 350 s, with the charging rate being as high as 166 µC s⁻¹. The fluctuation of current during the cycle is similar to that when charging the capacitor using a potentiometer (Supplementary Fig. 19). Together with a power management system, the hydrogel generator can collect mechanical energy and charge a 2 F capacitor, which then powered a LED, a thermometer, and a liquid crystal display (Supplementary Fig. 20 and Supplementary Fig. 21). An insole integrated with the hydrogel generator was designed to harvest mechanical energy from walking and store the charge in a capacitor (Supplementary Fig. 20). Furthermore, the hydrogel generator has outstanding durability, showing no significant degradation of output over 7500 compression cycles (Fig. 3e). This durability originates from the excellent elasticity of the device during loading cycles (Supplementary Fig. 22). The device can also be scaled up, achieving a peak current approaching 13 mA with the device area increased to 29 cm² (Supplementary Fig. 23). In addition, it is not surprising that the device can keep stable output even in a high-humidity environment (Supplementary Fig. 25), which would be otherwise difficult for TENG. Due to the physical

cross-linking of PVA, the hydrogel part can be recycled for disused devices (Supplementary Fig. 26).

Compared with state-of-the-art mechanoelectrical energy converters, including semiconductor direct-current triboelectric nanogenerator (SDC-TENG)[41], magnetoelastic generator[42], TENG[37–40,43], and PENG[44], this mechanoionic hydrogel generator is advantageous in terms of internal resistance, current density, transferred charge density, and device softness (Fig. 3f and Supplementary Table 1). In contrast to EMG and TENG, our hydrogel generator is especially suitable for harvesting very-low-frequency mechanical energy (Fig. 3g and Supplementary Table 1), and the current output increases with decreasing frequency[8]. Furthermore, considering the mechanical softness and biocompatibility of the hydrogel generator, it is promising for applications in soft robotics and biomedical devices[27].

To demonstrate its utility in biomedical applications, we used the hydrogel generator as the power source for an integrated drug-releasing system. A polypyrrole (PPy) layer doped with dopamine and an antibiotic (cefazolin sodium) was electrodeposited onto a CC

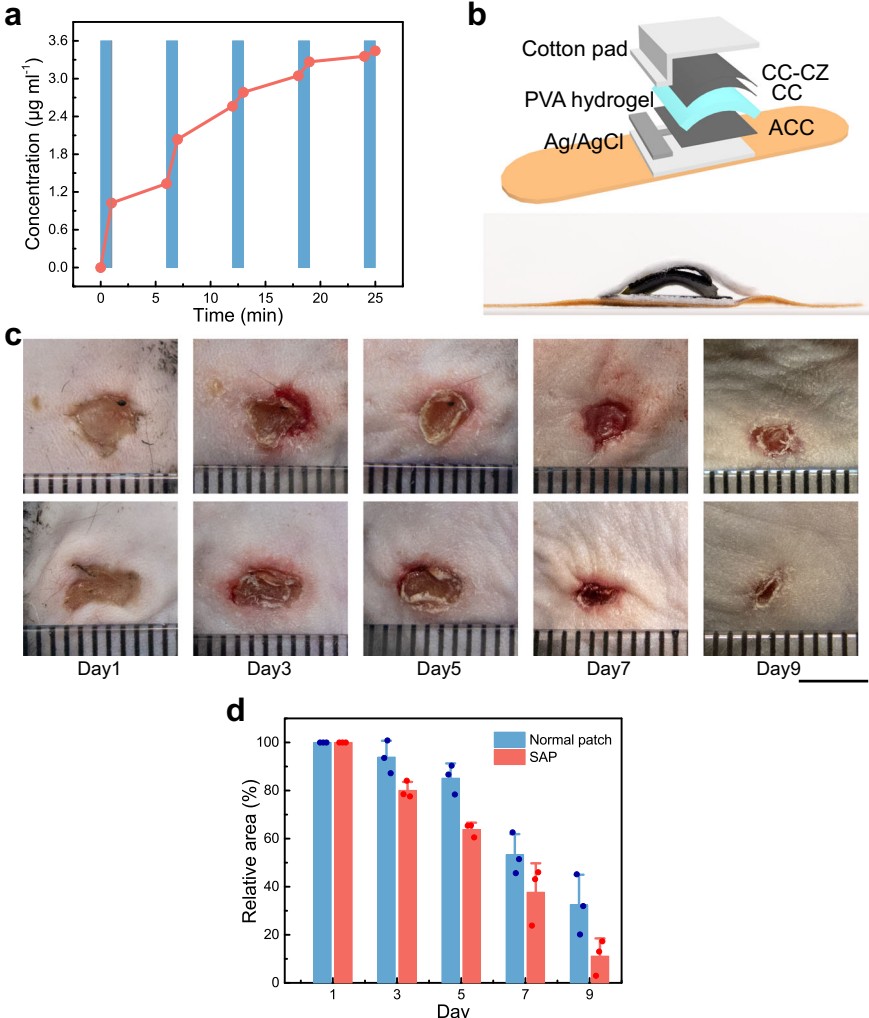

**Fig. 4 | Application in a drug releasing device. a** In vitro demonstration of electrically controlled drug release with "on-off" cycles driven by the hydrogel generator. The "on" state (blue) represents tapping the device to provide output current. **b** Schematic (upper) and photography (bottom) of the self-powered drug-releasing patch (see details and results of the animal experiment in Methods and Supplementary Fig. 29, 30). **c** Photographs of the wound during the healing process with a standard patch (upper) and the SAP (lower). Scale bar: 5 mm. **d** The relative area of wounds as a function of healing time. On the last day, the wound area covered by the normal patch and SAP was reduced to 32% and 11% of the initial wound area, respectively. Data are reported as their mean value ± SDs from $n = 3$ independent samples. Source data are provided as a Source Data file.

(Supplementary Fig. 27 and Supplementary Fig. 28). As illustrated in Supplementary Fig. 29, a two-electrode system was employed, in which the drug-loaded CC was the working electrode, a Pt foil was the counter electrode, and 10 ml PBS buffer was the electrolyte. Under periodic tapping (tapping for 1 min and resting for 5 min), the generated current transforms PPy from the oxidized state to the reduced state, releasing the negatively charged cefazolin from PPy to PBS solution[45]. The concentration of the drug was calibrated by UV-VIS spectroscopy[46] (Supplementary Fig. 30). Due to the high current output of the generator, the drug release rate when tapping the device is much higher than that without tapping (Fig. 4a), suggesting a design for electromechanically controlled release of drugs.

This device was further demonstrated in an in vivo application as a self-powered anti-bacterial patch (SAP) with sustained release of antibiotics, which is useful for the treatment of wound infection. Figure 4b presents an exploded view schematic image and a photograph of SAP, which includes an adhesive bandage substrate, Ag/AgCl reference electrode, ACC electrode, arched PVA hydrogel electrolyte, CC electrode, cefazolin modified carbon cloth (CC-CZ) patch, and wet cotton immersed in PBS for drug diffusion and protection. The SAP was

employed to promote wound healing in infected mice. After anesthesia and cleaning the operative area, we generated two symmetrical full-thickness skin incisions in the dorsal flank region of mouse and then infected with Staphylococcus aureus (S. aureus). Subsequently, the SAP was fixed to cover the wound on the right side, while a commercial adhesive bandage was fixed to cover the other wound as control (Supplementary Fig. 31). Movements of the mouse induce stretching of the bandages, which deform the arched hydrogel and change the physical contact between hydrogel layer and the working electrode. The generated current flows from ACC through the silver/silver chloride, the PBS-containing cotton cloth, to the drug-loaded CC-CZ and back to CC, as described in Supplementary Fig. 32. The sizes of the wounds were recorded every two days to evaluate the treatment efficiency, as presented in Fig. 4c, d. Evidently, the area of the wound covered by SAP is smaller than the area of the other wound covered by commercial bandage during the 9-day healing period. On the last day, the wound area covered by the SAP and regular patch was reduced to 13% and 45% of the initial wound area, respectively. The significant improvement of the wound healing by SAP indicates effective drug release driven by mechanoelectrical energy

conversion. Indeed, the biocompatibility, high current output, low voltage, and low impedance of the hydrogel generator could enable diverse applications in controlled drug-delivery as the chemistry and dosage can be further engineered.

## Discussion

In summary, we have realized a high current hydrogel generator based on amplified mechanoionic effects. Engineered structural and chemical asymmetry was optimized in synergy to enhance the net ionic current generation. The excellent outputs of current and charge transfer are advantageous for biomedical applications, as exemplified by a self-powered drug-releasing patch for wound healing. This ionic current generation mechanism is naturally compatible with living organisms[47] and living hydrogels[48], where ions are the dominant charge/information/energy carriers. On the other hand, hydrogel materials can be further engineered with additional properties, such as bioactivity, biodegradability, bioadhesion, and other characteristics useful in advanced bio-interfaces. Considering these attributes, this design of hydrogel mechanoionic generator will enable many promising applications in soft electronics, biomedical devices, etc.

## Methods

### Activated carbon cloth (ACC)

A reported method was used to fabricate ACC[28]. In brief, a piece of CC was cleaned with acetone, isopropyl alcohol (IPA), and de-ionized water (DI water) before activation. The electrochemical oxidative process was carried out in a two-electrode system in 0.1 M $(NH_4)_2SO_4$ aqueous solution using a Pt electrode as the counter electrode. A voltage of 10 V was applied for 15 min. Then the electrochemical reduction process was carried out in a three-electrode system in 1 M $NH_4Cl$ aqueous solution with the Pt and Ag/AgCl electrodes serving as the counter and reference electrodes, respectively. The oxidized CC was reduced at −1.2 V for 30 min. After cleaning and drying, the as-fabricated ACC was collected for further use.

### CC-PVA counter electrode

First, 15 wt% PVA (alcoholysis degree: 99 + %, molecular mass: 146,000–186,000) solution was poured into a 3D-printed PTFE mold, frozen at −18 °C for 8 h, and thawed at 25 °C for 3 h. This process was repeated four times to improve mechanical performance. Then, a piece of cleaned CC was immersed into 10 wt% PVA solution and placed onto the back side of the demolded PVA hydrogel. After that, another two freezing-thawing cycles were conducted. The integrated CC electrode-PVA hydrogel was kept in the electrolyte solution (4 M LiCl) for 1 day. Here, high-concentration PVA solution (15 wt%) was utilized to prepare the hydrogel matrix to ensure excellent mechanical properties (high modulus, high resilience) after multiple freezing-thawing processes and its easiness to be completely detached from the mold. However, 15 wt% PVA solution is too viscous and cannot penetrate the CC electrode. Therefore, at the CC electrode surface, low-concentration solution (10 wt%) was used in order to integrate the PVA hydrogel with the CC electrode.

### PI/Au and steel electrodes

PI/Au electrode: A piece of PI film (thickness: 50 μm) was cleaned with acetone, IPA, and DI water. Then a bilayer of chromium (Cr, 5 nm) and gold (Au, 200 nm) was sputtered (Denton desktop pro sputter system) on the PI film.

Steel electrode: A piece of stainless-steel foil (50 μm) was cleaned with sandpaper (7000 mesh), acetone, IPA, and DI water.

The oxygen plasma treatment (40 sccm oxygen flow, 100 W, and 30 s) was carried out by an RIE etcher (Tailong Electronics).

PI/Au-PVA counter electrode: A piece of PI/Au electrode without plasma treatment was used, and other steps were the same as the mentioned CC-PVA electrodes.

### PEI modified ACC, and the removal process

After cleaning, a piece of ACC was immersed into 25 wt% PEI solution (600 MW, Aladdin Company). PEI with positively charged groups was adsorbed onto the surface of ACC during the 24 h immersion. Then the PEI-modified ACC, named ACC-PEI, was completed by drying it in an oven at 85 °C.

After measurement, this electrode was rinsed with DI water for 2 min and dried again. This step aims to reduce the content of PEI. After that, the electrode was tested again. Afterward, the ACC was immersed in DI water for 2 days at 85 °C to remove most of the adsorbed PEI. The as-fabricated ACC was tested after drying.

For ACC-PEI as the counter electrode, we adopted a slightly different method for modification with better chemical stability[49]. After cleaning, a piece of CC was placed in 65% HNO3 at 90 °C for 3 h. Then this CC was placed in DCM solution with 10% PEI at 50 °C for 2 h.

### Drug-delivery patch (CC-CZ)

A three-electrode system was constructed, in which a piece of cleaned CC, a Pt electrode, and an Ag/AgCl electrode serve as the working electrode, the counter electrode, and the reference electrode, respectively. A constant voltage of 0.8 V was applied to the system for 300 s with the electrolyte solution containing 0.1 M pyrrole, 0.01 M dopamine hydrochloride, and 0.01 M cefazolin sodium. After that, a layer of chloride-doped polypyrrole was deposited via electrodeposition (0.8 V, 60 s) to slow down the natural release of the drug (the electrolyte contains 0.1 M pyrrole, 0.01 M dopamine hydrochloride, and 0.1 M sodium chloride). The as-prepared patch CC-CZ was immersed in PBS buffer and DI water to remove the residual reagents.

### Self-powered anti-bacteria patch

Fabrication of the arched CC/PVA structure: a flattened CC/PVA electrode was first obtained by casting the PVA solution onto CC and freezing. After the first freezing-thawing cycle, the CC/PVA electrode was put into a beaker to introduce the bending deformation. Another three freezing-thawing cycles were applied to fabricate the arched CC/PVA electrode.

Fabrication of the SAP: a commercial adhesive bandage was employed as the substrate. Then the Ag/AgCl circuit was printed by coating a layer of Ag/AgCl ink with a mask on the central cotton pad, followed by annealing. The ACC electrode, arched CC/PVA electrode, drug-delivery patch, and another cotton pad was put onto the band-aid layer-by-layer. Finally, the upper cotton pad was fixed with waterproof medical tape.

### Characterization of materials

SEM and EDS mapping of ACC were characterized by Hitachi S-3400N. ESCALab 250 XI was used for the X-ray photoelectron spectrometry. SEM images of the drug-delivery patch (CC-CZ) were obtained by Hitachi S-4800. The contact angle was tested by Attention Theta, Biolin Scientific. FTIR spectrum was tested by Thermo Scientific Nicolet™ iS™ 5 FTIR Spectrometer. EIS data was obtained by an electrochemistry workstation Autolab PGSTAT 302 N. The ion conductivity and ion diffusion rate are calculated from the EIS data using reported methods[50–54].

### Characterization of the mechanical-electrical responses

A mechanical tester (Zwick Roel) was used to control the movement of the working electrode at 24–27 °C. Testing parameters are listed in Supplementary Table 3. Electrical signals were measured by an electrochemistry workstation Autolab PGSTAT 302 N. Unless otherwise specified, measurements were repeated for five cycles for each experiment.

The power density (P) was calculated by

$$P = I^2/(R \times S) \tag{1}$$

with $I$, $R$, $S$ being the recorded current, the resistance of loading, and the area of the device, respectively. The transferred charge amount ($Q$) was calculated by

$$Q = \int I dt \tag{2}$$

where $I$ is the current and $t$ is the time. Before testing, the surface of the hydrogel and the counter electrode were wiped dry.

## Finite element analysis

A commercial FEA software COMSOL was used for the simulation. A pyramid hydrogel supported with a rectangular base and a cubic hydrogel were constructed (Fig. 1c and Supplementary Table 2). We modeled the hydrogel as an elastic solid and introduced a stainless-steel plate to compress the hydrogels in the $z$ direction shown in Fig. 1c. A fixed boundary condition was applied to the bottom of the hydrogels. We calculated the principal strain $e_{zz}$ which significantly contributes to the mechanoionic effect.

The strain $e_{zz}$ in the central cross section of each sample perpendicular to $y$ direction was plotted. We determined the average strain gradient $\frac{de_{zz}}{dz}$ near the top surface (The depth from the surface is 10% of the height of hydrogel, as shown in Supplementary Fig. 2. The boundary is indicated as a dashed line.). After converting the strain results from the heatmap (Supplementary Fig. 2a, c) to the contour map (Supplementary Fig. 2b, d), the density of the contours is calculated at six equally spaced points in the electrode-electrolyte interface.

## DFT calculation

All spin-polarized first-principle computations were performed under the framework of DFT. Perdew-Burke-Ernzerhof function with a generalized gradient approximation was introduced to describe the exchange and correlation effect of electrons. The projector augmented wave method was employed to describe the electron-ion interaction. The kinetic energy cutoff for plane wave was set to 400 eV. The Brillouin zone was sampled by a $3 \times 3 \times 1$ Gamma grid. The criterion of structure relaxation was set to $10^{-4}$ eV for total energy and 0.02 eV/Å for the force of each atom. DFT-D3 method was employed for the correction of vdWs force. A $4 \times 4$ single-layer graphite supercell containing 32 atoms was employed to model the pristine CC. The oxidized CC was simulated by adding an oxygen atom on the above single-layer graphite, forming a C-O-C epoxy group.

The adsorption energy $G_a$ of $Li^+$ and $Cl^-$ ions adsorbed on the pristine/oxidized CC is given by:

$$G_a = -G_{ions@substrate} + G_{substrate} + G_{ions/water} \tag{3}$$

where $G_{ions@substrate}$, $G_{substrate}$ and $G_{ions/water}$ denote the Gibbs free energy of the pristine/oxidized single-layer graphite with water-coordinated ions, pristine/oxidized single-layer graphite and ions with water layer, respectively.

## Electrical-responsive drug release in vitro

In order to achieve a calibration curve, a series of PBS buffers containing different concentrations of the drug (cefazolin sodium) were prepared. A UV-Vis spectrometer defined the relationship between the absorbance and the drug concentration at 272 nm, which is the characteristic absorbance band of cefazolin sodium. Therefore, a standard calibration curve of cefazolin sodium was plotted, as shown in Supplementary Fig. 30, with $R^2 = 0.99744$.

A piece of CC-CZ (1.2 cm × 1.4 cm), Pt electrode, and 10 mL PBS buffer formed a two-electrode system. The Pt electrode was connected to a piece of ACC. The CC-CZ was fixed by a Pt clip and connected to a CC-PVA electrode, as shown in Supplementary Fig. 29. Hand-tapping

the ACC placed on the CC-PVA electrode generated a current pulse and released the drug. After 1 min tapping, 2 mL of the PBS buffer was moved to a quartz cuvette and tested by the UV-Vis spectrometer. Then the PBS buffer was poured back into the two-electrode system and kept for 5 min before the next test. This process was repeated 4 times.

## Electrical-responsive drug release in vivo

All experimental animal procedures were performed in accordance with ethical approval by the Animal Research Ethics Sub-Committee of City University of Hong Kong. Balb/c mouse (female, 6–8 weeks, 20–25 g) were used to perform this experiment and were group-housed until the start of this experiment. The environment was maintained at 24–27 °C and 50–80% humidity Lights were kept on a 12-h on/12-h off cycle. All experiments were performed during the light time.

After anesthesia (pentobarbital, 1.5 mg/100 g, intraperitoneal injection), an electric razor and hair removal cream was applied to shave the hair from a dorsal flank region of the mice. Two symmetrical skin incisions were generated on both sides of each mouse. Then the wounds were infected by 200 μL Staphylococcus aureus (S. aureus) with an initial density of $10^7 CFU \, mL^{-1}$. After that, all wounds were covered by transparent waterproof wound dressing bandage (Nexcare, 3 M). Mice were kept in cages overnight.

The initial areas of wounds were recorded the day after surgery. After anesthesia, a camera (EOS R5, Canon) was used to record the areas of wounds. Then the self-powered anti-bacteria band-aid covered a wound on one side of the mice. Before using, the upper cotton pad was wetted with sterile PBS buffer. A normal band-aid covered the other wound. This process was repeated every two days until the ninth day. The areas of wounds were measured using ImageJ software. After experiments, mice were euthanized by carbon dioxide asphyxiation.

## Reporting summary

Further information on research design is available in the Nature Portfolio Reporting Summary linked to this article.

# Data availability

Additional data are available from the corresponding author upon request. The data supporting the findings of this study are available within the article and its Supplementary Information, as well as Source Data. Source data are provided with this paper.

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

## Acknowledgements

L.X. acknowledges funding support from Research Grants Council (RGC), University Grants Committee (UGC) (Project 17200722 and 17200320), and Environment and Conservation Fund (project 125/2021). This work was also supported by the Health@InnoHK program of the Innovation and Technology Commission of the Hong Kong SAR Government. The authors are thankful for the helpful discussion with Prof. Zhonglin Wang from Georgia Institute of Technology, Dr. Wei Xu from Zhejiang Lab, Mr. Huimin He and Ms. Sijia Wang from The University of Hong Kong, and the technical support from Dr. He Zhang, Mr. Shenghua Zhou, and Mr. Yiyang Fei from The University of Hong Kong.

## Author contributions

Hongzhen Liu conceived the idea and completed the experimental investigation. Xianglin Ji, Hongzhen Liu, and Peng Shi designed and completed the animal experiment. Zihao Guo contributed to the figure presentation. Hongzhen Liu and Xi Wei contributed to the FEA calculation. Jinchen Fan provided critical insights for the working mechanism of the device. Feng Gong contributed to the DFT calculation. Hongzhen Liu and Xiong Pu developed the working mechanisms of the devices. Hongzhen Liu, Xiong Pu, and Lizhi Xu completed the manuscript. Lizhi Xu supervised the whole project.

## Competing interests

The authors declare no competing interests.
