## [Peer review file · Nature Communications]

REVIEWER COMMENTS

Reviewer #1 (Remarks to the Author):

Liu et al. reported an interesting study relevant to PVA hydrogel-based mechanoionic current generator. Owing to the design of pyramid-structured hydrogels and AAC electrode, the generator outputs a high compression-current conversion efficiency. I appreciate the novelty of this work towards to design of hydrogel-based generator and its high efficiency, however, I am afraid that the unique of hydrogel material and generator application were unclarified or unconvincing. Overall, the idea in this work is interesting, but biological application and materials were vaguely described and lack of supporting tests.

Major revision should be required.

1. Why did you chose PVA hydrogels? The authors should compare and investigate poly(acrylic acid), zwitterionic, or other polyelectrolyte hydrogels that may improve the efficiency of this generator.
2. What is the alcoholysis degree and molecular mass of PVA? Why did you choose two concentrations of PVA solution and different frozen-thaw process? No any examination in the manuscript.
3. Why did Cl⁻ diffuse at a higher rate than Li⁺ in hydrogel?
4. This hydrogel-based generator required the high cyclic compressibility. I am afraid that the physical crosslinking PVA hydrogel may be not stable in cyclic compression tests. How did you solve the dehydration of PVA, especially in applications?
4. In vivo wound healing application was confusing and unconvincing. How to make the device stretch repeatably to generate electricity? What does mean 'normal patch', were there antibiotics in it? If not, how can the infected wound be healing so quickly? The comparison group should be commercial wound dressing. The wound area in two groups seemed to be quite different at Day 1. While there was only one sentence that described this test in the manuscript.
5. 'Profiles of the generated short-circuit current, open-circuit voltage and applied pressure' was repeatedly described in caption of Fig. 1d and e
6. English writting should be improved.

Reviewer #2 (Remarks to the Author):

This manuscript presents an innovative hydrogel generator utilizing the piezoionic effect, which can generate high peak currents and transferred charges. Moreover, the authors demonstrate the potential application of this generator in self-powered systems. Before publication in Nature Communications, the authors are suggested to address the following issues to enhance the manuscript quality:

1. In Figure 1A, the schematic depicts the pyramid shape of the patch. However, the photo provided does not clearly show the structure. The authors should provide an additional photo from a different angle that offers a clearer view of the patch's structure.
2. In Figure S1, a comparison was made between the cubic and pyramid designs. However, the strain distribution of the cubic hydrogel is confusing as the top and bottom surfaces exhibit symmetrical strain, and the highest strain is observed on the sides rather than the center. To address this, the authors should clarify the orientation of the cubic PVA gel and provide labeling for the direction of the force. Additionally, the authors should specify the thickness of the cubic hydrogel, as thinner hydrogels can lead to higher strain under the same pressure. Furthermore, it is difficult to determine if the cubic hydrogel and pyramid hydrogel were subjected to loading with the same shape and area.

3. In terms of the device structure, when the ACC initially contacts the hydrogel, cations are expected to be absorbed due to the higher absorption energy of the ACC. However, this raises the question of how the cations and anions can distribute uniformly at the starting point or after recovery.
4. Regarding the material selection for the device, it is recommended that the authors should explain why they did not choose an electrode with an affinity for anions. Additionally, the reviewer wonders why the authors did not explore the possibility of using two different ACCs to generate a larger flux.
5. It is noted that the authors did not include several plots that contain crucial characterization information for a piezoionic generator. Specifically, the plots "Ion flux vs. Time," "Ionic conductivity vs. electrolyte concentration," and "Diffusion Coefficient vs. electrolyte concentration" were absent from the manuscript.
6. Regarding the PEI experiment, it may not be necessary to include it as a main figure since Figure 2a has already demonstrated the significance of ion selectivity. Moreover, the schematics in Figure 2e are identical, which should be a typo.
7. To ensure a comprehensive evaluation of the device's stability, it is recommended that the authors provide data characterizing the performance after subjecting it to long-term rinsing in water or exposure to air. Considering the potential dehydration of the hydrogel over time, which could lead to a decrease in performance, this information is crucial for understanding the device's long-term viability.
8. To optimize the device performance, it is crucial to consider the concentration of the LiCl solution, activation time of the ACC electrodes, and applied pressure. It is recommended that the authors explore whether there are specific threshold values for the concentration and activation time that yield optimal performance. Additionally, the authors should address the reason behind the saturation of current with increasing pressure.
9. In Figure S14, the application did not show the electrical connection to the mechanoionic generator. The authors should add the schematics for the application circuit and realistic photos of the generator connected to the application circuit.
10. Figure 4 currently lacks sufficient information. The authors are suggested to include additional data to enrich its content, such as the animal experiment results or other applications.
11. In figure S22, the number of calibration data points needs to be increased. The author should provide more data points uniformly in all ranges.
12. Instead of solely comparing its performance with other technologies, the authors should focus on discussing the feasibility and practicality of this approach in the application for drug delivery.

Reviewer #3 (Remarks to the Author):

The authors describe their interesting design of mechanoionic generators by using hydrogels in conjunction with asymmetric electrode structures. The produced current is very high in comparison to literatures. They carefully characterised their devices in terms of performance and compared it with the state-of-the-art literature, and finally demonstrated applications in controlled drug delivery. I recommend publication after minor revision by addressing the following issues:

- (1) I wonder the generality of their system. how crucial is PVA hydrogel? would it work for other hydrogel also?
- (2) Figure 3e demonstrated the high durability of their device. nevertheless, I expect PVA hydrogel would be sticky and may adhere to the ACC electrode surfaces with repeating loads or high pressure. Also, the ACC surface may be pretty rough. Would pyramid tips stuck into the electrode surfaces after repeated pressing? some explanation on the soft/hard interface may be helpful by referring to the related literature (<https://doi.org/10.1002/adma.201902278>).
- (3) what is the estimated mechanical-electricity conversion efficiency?

Dear Reviewers,

Thank you very much for your consideration of this work and the insightful comments. The comments and suggestions are helpful to improve the paper quality. According to the insightful suggestions, the manuscript has been revised carefully, and the changes are marked in **yellow**. The point-to-point response to each reviewer is given below.

Reviewer #1

1. Why did you chose PVA hydrogels? The authors should compare and investigate poly(acrylic acid), zwitterionic, or other polyelectrolyte hydrogels that may improve the efficiency of this generator.

Response:

We sincerely appreciate your kind comments. The selection of PVA hydrogel for demonstration purposes is based on several factors. (1) Firstly, PVA hydrogel has been studied for many years and can be easily obtained with highly consistent mechanical properties through simple freeze-thaw cycles. This allows for better control of variables during the experimental process. (2) Secondly, by increasing the solid content and the number of freeze-thaw cycles, PVA hydrogel with higher mechanical properties can be obtained. This is beneficial for the demolding process of the hydrogel and helps maintain its resilience, morphology, structure, and stable mechanical properties during subsequent compression-separation testing, enabling it to withstand higher pressures. (3) Additionally, PVA hydrogel obtained through freeze-thaw cycles can be conveniently assembled, facilitating seamless integration with electrode carbon cloth. (4) Finally, the relatively simple and stable chemical structure of PVA is advantageous for analyzing the device properties.

Based on the reviewer's suggestions, we tested the electrical outputs using two different hydrogels as substitutes for PVA, including PAAM and GelMa. As shown in Fig. S12, with 1M LiCl as the electrolyte, GelMa hydrogel exhibited the highest output current under the same test conditions (same pressure, strain rate of 16 mm/s, and time interval of 10.8 s), followed by PVA hydrogel. Both of them achieved milliampere-level output currents. PAAM only showed limited output currents. In fact, PAAM hydrogels exhibit significantly higher viscosity, which indicate additional interfacial interaction with ACC that could compromise the ion-anchoring effect. The larger output of GelMa hydrogel is probably because it is softer and therefore larger ACC-hydrogel interface area at compressed state than that of the PVA hydrogel.

We understand that the polyelectrolyte hydrogel could possibly improve or weaken the mechanoionic effect, as documented in literature (*Science* 2022, **376**, 502-507; *Macromolecules* 2010, **43**, 5814-5819; *Macromolecules* 2010, **43**, 511-517.) Polycation electrolyte hydrogel could possibly enhance the electrical outputs due to migration of single-sign anions, yielding a streaming current (*Macromolecules* 2010, **43**, 5814-5819; *Macromolecules* 2010, **43**, 511-517.) Here, we didn't focus on the effect of different polymer networks, but rather the electrode-hydrogel interfaces. As also demonstrated by the low outputs of PAAM hydrogel, the interface is of great importance.

We added a figure in Supporting Information (Fig. S12) and added a caption to provide the relevant information.

Revision:

Main text:

Results

...When the PVA hydrogel is replaced with a gelatin methacryloyl (GelMA) hydrogel, this phenomenon can still be observed, demonstrating the generality of the proposed mechanism (Fig. S12). When using PAAM for the hydrogel layer, however, the output current is significantly lower. In fact, PAAM hydrogel exhibit much greater viscosity than PVA and GelMA hydrogels. The distinct interfacial interactions may affect the surface functionality of ACC, which is worth future investigations³⁴.

Supplementary Figures:

Fig. S12 Current outputs of different hydrogels. Three different hydrogels were tested, i.e. PVA, PAAM, and GelMa hydrogels. All the hydrogels were soaked in 1M LiCl solution and air-dried before testing. The strain rate (16 mm/s) and interval time (10.8 s) were the same for all tests. Both PVA and GelMa hydrogels achieved milliampere-level output currents; PAAM hydrogel only showed microampere-scale currents. In fact, GelMa and PVA hydrogels do not exhibit high adhesion to ACC; in contrast, PAAM hydrogel, when taken out from the salt solution and air-dried for a period, exhibited high adhesion to ACC. The right figure (b) shows the mass change of the ACC electrode before and after contacting the PAAM and PVA hydrogel, respectively. After being pressed against the PAAM hydrogel, the ACC surface is coated with a layer of hydrogel or salt solution, which may affect the surface functionality. The larger output of GelMa hydrogel is probably because it is softer and hence a larger ACC-hydrogel interface area at compressed state than that of the PVA hydrogel. Therefore, the PVA hydrogel was finally selected based on its following merits: (1) PVA hydrogel can be easily obtained with highly consistent mechanical properties through simple freeze-thaw cycles. This allows for better control of variables during the experimental process. (2) by increasing the solid content and the number of freeze-thaw cycles, PVA hydrogel with higher mechanical properties can be obtained. This is beneficial for the demolding process of the hydrogel and helps maintain its resilience, morphology, structure, and stable mechanical properties during subsequent

compression-separation testing, enabling it to withstand higher pressures. (3) PVA hydrogel obtained through freeze-thaw cycles can be conveniently assembled, facilitating seamless integration with electrode carbon cloth. (4) the relatively simple and stable chemical structure of PVA is advantageous for analyzing the device performance.

2. What is the alcoholysis degree and molecular mass of PVA? Why did you choose two concentrations of PVA solution and different frozen-thaw process? No any examination in the manuscript.

Response:

We are very grateful for the reviewer's careful reviewing for our work in details. The alcoholysis degree and molecular mass of PVA are 99+% and 146,000-186,000, respectively. We indeed used two PVA solutions with different concentrations. High-concentration PVA solution (15 wt%) was used to prepare the pyramid hydrogel matrix; while low-concentration PVA solution (10 wt%) was used to ensure the intimate hydrogel-CC interfaces. PVA hydrogels using 15wt% concentration can achieve excellent mechanical properties (high modulus, high resilience) after multiple freezing-thawing processes, allowing it to be completely and easily detached from the mold. The excellent mechanical properties are also important in the compressing-separating tests. However, 15 wt% PVA solution is too viscous and cannot penetrate into the CC electrode. Therefore, at the CC electrode surface, we used a low-concentration solution (10 wt%) in order to integrate the PVA hydrogel with carbon cloth. We first prepared the pyramid hydrogel, and demold it. Then, put the CC electrode on the backside and soak the low-concentration PVA solution. After that, the following freeze-thaw cycles were carried out.

We have modified the CC-PVA counter electrode part in Method section in the manuscript, adding more information about the preparing procedures.

Revision:

Main text:

Methods

First, 15 wt% PVA (alcoholysis degree: 99+%, molecular mass: 146,000-186,000) solution was poured into a 3D-printed PTFE mold, frozen at -18 °C for 8 h, and thawed at 25 °C for 3 h. This process was repeated four times to improve mechanical performance. Then, a piece of cleaned CC was immersed into 10 wt% PVA solution and placed onto the back side of the demolded PVA hydrogel. After that, another two freezing-thawing cycles were conducted. The integrated CC electrode-PVA hydrogel was kept in the electrolyte solution (4M LiCl) for 1 day. Here, high-concentration PVA solution (15 wt%) was utilized to prepare the hydrogel matrix to ensure excellent mechanical properties (high modulus, high resilience) after multiple freezing-thawing processes and its easiness to be completely detached from the mold. However, 15 wt% PVA solution is too viscous and cannot penetrate into the CC electrode. Therefore, at

the CC electrode surface, low-concentration solution (10 wt%) was used in order to integrate the PVA hydrogel with the CC electrode.

3. Why did Cl⁻ diffuse at a higher rate than Li⁺ in hydrogel?

Response:

We are very grateful for the reviewer's careful reviewing and kind comments. In aqueous solutions with concentrations not exceeding 16M, the hydrated chloride ions have a smaller radius than hydrated lithium ions and exhibit higher diffusion rates (Chem. Soc. Rev., 2015, 44, 7484-7539). In hydrogels, the ion-polymer network interactions could make it slightly complicated. Polymer chain with stronger interaction with cation will certainly lower the cation transfer. It has been reported that highly crystalline PVA exhibits coordination interactions with alkali metal cations, further reducing the mobility of the cations (Sci. Adv., 2021, 7, 48, eabi7233). Therefore, in highly crystalline hydrogels, chloride ions diffuse faster.

We have added the relevant references in the 3rd line on page 7 of the main text.

Revision:

Main text:

Results

...It is known that Cl⁻ anions diffuse at a higher rate than Li⁺ cations in aqueous solution³¹, and this difference in diffusivity is even greater in PVA hydrogels³²...

4. In vivo wound healing application was confusing and unconvincing. How to make the device stretch repeatably to generate electricity? What does mean 'normal patch', were there antibiotics in it? If not, how can the infected wound be healing so quickly? The comparison group should be commercial wound dressing. The wound area in two groups seemed to be quite different at Day 1. While there was only one sentence that described this test in the manuscript.

Response:

We are very grateful for the reviewer's careful review and kind comments. During movement, respiration, and eating, the abdomen of the mouse undergoes repeated contractions and relaxations, causing the patch to stretch and restore its shape. When the arched patch is flattened, the arched PVA hydrogel gets compressed against the ACC. As the patch regains its original shape, the PVA separates from the ACC, and this process repeats to generate electrical energy. The term "normal patch" refers to a standard band-aid without any additional treatment. The specific process of wound healing varies significantly depending on the mouse itself. Mice in good physical condition tend to have faster wound healing rates. To account for individual differences, we conducted three sets of experiments to eliminate discrepancies caused by varying physiques. Regarding regular band-aids, the information we gathered indicates that the inner layer typically comprises non-woven fabric and lacks antibacterial ingredients.

Commercial dressings from 3M are used to wrap and secure band-aids and wounds. Despite variations in wound shapes, measurements revealed that the wound areas on both sides were nearly equal on the first day, measuring 15.631 (for the normal patch) and 15.632 (for SAP), respectively.

We have revised the last paragraph of the Results section and Fig. 4 to add details about the animal experiments.

Revision:

Main text:

Results

To demonstrate its utility in biomedical applications, we used the hydrogel generator as the power source for an integrated drug-releasing system. A polypyrrole (PPy) layer doped with dopamine and an antibiotic (cefazolin sodium) was electrodeposited onto a carbon cloth (Fig. S27 and Fig. S28). As illustrated in Fig. S29, a two-electrode system was employed, in which the drug-loaded carbon cloth was the working electrode, a Pt foil was the counter electrode, and 10 ml PBS buffer was the electrolyte. Under periodic tapping (tapping for 1 min and resting for 5 min), the generated current transforms PPy from the oxidized state to the reduced state, releasing the negatively charged cefazolin from PPy to PBS solution. The concentration of the drug was calibrated by UV-VIS spectroscopy (Fig. S30). Due to the high current output of the generator, the drug release rate when tapping the device is much higher than that without tapping (Fig. 4c), suggesting a design for electromechanically controlled release of drugs.

This device was further demonstrated in an in vivo application as a self-powered anti-bacterial patch (SAP) with sustained release of antibiotics, which is useful for the treatment of wound infection. Fig. 4d presents an exploded view schematic image and a photograph of SAP, which includes an adhesive bandage substrate, Ag/AgCl reference electrode, ACC electrode, arched PVA hydrogel electrolyte, CC electrode, drug-loaded CC-CZ patch, and wet cotton immersed in PBS for drug diffusion and protection. The SAP was employed to promote wound healing in infected mice. After anesthesia and cleaning the operative area, we generated two symmetrical full-thickness skin incisions in the dorsal flank region of mouse and then infected with *Staphylococcus aureus* (*S. aureus*). Subsequently, the SAP was fixed to cover the wound on the right side, while a commercial adhesive bandage was fixed to cover the other wound as the control (Fig. S31). Movements of the mouse induce stretching of the bandages, which deform the arched hydrogel and change the physical contact between hydrogel layer and the working electrode. The generated current flows from ACC through the silver/silver chloride, the PBS-containing cotton cloth, to the drug-loaded CC-CZ and back to CC, as described in Fig. S32. The sizes of the wounds were recorded every two days to evaluate the treatment efficiency, as presented in Fig. 4e and 4f. Evidently, the area of the wound covered by SAP is smaller than the area of the other wound covered by commercial bandage during the nine-day healing period. On the last day, the wound area covered by the SAP and regular patch was reduced to 13% and 45% of the initial wound area, respectively. The significant improvement of the wound healing by SAP indicates effective drug release driven by mechano-electrical energy conversion.

5. 'Profiles of the generated short-circuit current, open-circuit voltage and applied pressure' was repeatedly described in caption of Fig. 1d and e

Response:

We are very grateful for the reviewer's careful reviewing and kind comments. We have deleted the sentence in caption of Fig. 1d.

Revision:

Main text:

Fig. 1 ...d, C1s XPS spectra for CC and ACC, indicating incorporated oxygen on ACC. e, Profiles of the generated short-circuit current, open-circuit voltage and applied pressure...

6. English writing should be improved.

Response:

We are very grateful for the reviewer's careful reviewing and kind comments. We have improved the English of the manuscript.

Reviewer #2 (Remarks to the Author):

1. In Figure 1A, the schematic depicts the pyramid shape of the patch. However, the photo provided does not clearly show the structure. The authors should provide an additional photo from a different angle that offers a clearer view of the patch's structure.

Response:

We are very grateful for the reviewer's careful review and kind comments. The updated photo was placed in Fig. S1.

Revision:

Supplementary Figures:

Fig. S1 Digital photo of a large area PVA hydrogel with pyramid structure.

2. In Figure S1, a comparison was made between the cubic and pyramid designs. However, the strain distribution of the cubic hydrogel is confusing as the top and bottom surfaces exhibit symmetrical strain, and the highest strain is observed on the sides rather than the center. To address this, the authors should clarify the orientation of the cubic PVA gel and provide labeling for the direction of the force. Additionally, the authors should specify the thickness of the cubic hydrogel, as thinner hydrogels can lead to higher strain under the same pressure. Furthermore, it is difficult to determine if the cubic hydrogel and pyramid hydrogel were subjected to loading with the same shape and area.

Response:

We are very grateful for the reviewer's careful review and kind comments. The specific dimension of the model was listed in Table S2. After a series of adjustments, we controlled the real pressure (applied total force/area) of the two structures at around 31 kPa (32 kPa for the square structure and 30 kPa for the pyramid structure). Here, the area for the square structure is calculated according to the initial value, due to the relatively low deformation of the square structure under compression force. For the pyramid structure, we used the area of the top of the model after compression deformation, because the geometry parameters of this model changed significantly under compression. Due to the different geometric structures of the two, the total force on the square structure will be greater. However, even in this case, the strain gradient of the pyramidal structure under stress is still higher. In addition, it can be seen from the cross-sectional view that under the square structure, the strain is symmetrically distributed, with high

around the edges and low in the middle. Since ions tend to diffuse from the stress concentration area to the non-concentration area, the symmetrical structure will cause the ion flows to cancel each other, reducing the output current. The strain gradient generated by the pyramid structure is from top to bottom, which is more conducive to generating directional ion flow.

We modified Fig. S2, adding the direction of the applied force and coordinate system, and also modified the caption.

Revision:

Supplementary Figures:

Fig. S2 FEA results of compressive strain in center sections of cubic PVA hydrogel (a for heatmap and b for contour map) and patterned PVA hydrogel (c for heatmap and d for contour map). The direction of the force is from top to bottom. For the pyramid structure, we used the area of the top of the model after compression deformation, because the geometry parameters of this model changed significantly under compression. Under similar pressure (32 kPa for a and 30 kPa for c), the strain contours clearly show that the hydrogel with pyramidal structure has significantly enhanced strain gradient. In addition, it can be seen from the cross-sectional view that in the square structure, the strain is symmetrically distributed, with high strain around the edges and low strain in the middle. Since ions tend to diffuse from the high-strain area to the low-strain area, the symmetrical structure may cause the ion flows to cancel each other, reducing the output current. The strain gradient generated by the pyramid structure is from top to bottom, which is more conducive to generating directional ion flow.

3. In terms of the device structure, when the ACC initially contacts the hydrogel, cations are expected to be absorbed due to the higher absorption energy of the ACC. However, this raises the question of how the cations and anions can distribute uniformly at the starting point or after recovery.

Response:

We are very grateful for the reviewer's careful reviewing and kind comments. Before the contact between ACC the hydrogel, it can be assumed that lithium ions and chloride ions are evenly distributed in hydrogel. Once the device undergoes the compression-separation process, the uniform ion distribution in the hydrogel will be disrupted, with chloride ions moving away from the interface and lithium ions approaching the ACC interface. As ions move slowly, especially in highly concentrated solutions, it takes a certain amount of time for the two ions to return to their initial state. If the interval time is too short, lithium ions and chloride ions will still be unevenly distributed, which will cause the current value to drop during the second test, as described in Fig. 3 and Fig. S17. This phenomenon also shows that the interval time has a significant impact on the output current value. Therefore, our devices will perform better at low frequencies and are more suitable for low-frequency mechanical energy conversion.

We have mentioned this phenomenon in the caption of Fig. S17 and the second paragraph of the part "Output performance and applications".

Revision:

Main text:

Results

... This phenomenon can be explained as the ions cannot restore the initial uniform distribution within the short recovery time...

Supplementary Figures:

Fig. S17 Output current under various recovery times. Longer recovery time (lower frequency) generates more stable and higher currents, which may be attributed to the transient distribution of ions.

4. Regarding the material selection for the device, it is recommended that the authors should explain why they did not choose an electrode with an affinity for anions. Additionally, the reviewer wonders why the authors did not explore the possibility of using two different ACCs to generate a larger flux.

Response:

We are very grateful for the reviewer's careful reviewing and kind comments. We chose ACC

with cation affinity mainly due to its facile preparation, reliable consistency, and high stability. As you can see, we also tried to modify ACC with PEI to obtain an electrode with anion affinity and performed related experiments (Fig. 2e). However, from a practical application perspective, using ACC is obviously a more stable and convenient way. Reducing the processing steps for ACC can effectively improve the consistency and stability of the prepared electrodes. The binding force of PEI on ACC is mainly based on electrostatic force, so it is easily to be peeled off. But your suggestion is indeed insightful to support the working mechanism. According to this comment, we used PEI-modified ACC as the counter electrode, ACC as the working electrode, and PVA hydrogel swelled in 1M LiCl as the electrolyte to conduct relevant experiments. The results indicate that this combination exhibits the highest output current, consistent with our hypothesized mechanism.

We have added Fig. S14 to describe this phenomenon.

Revision:

Main text:

Results

Furthermore, by employing ACC-PEI as the counter electrode and ACC as the working electrode, we achieved a higher output current compared to that with CC as the counter electrode (Fig. S14).

Methods

For ACC-PEI as the counter electrode, we adopted a slightly different method for modification with better chemical stability⁴⁹. After cleaning, a piece of carbon cloth was placed in 65% HNO₃ at 90°C for 3h. Then this carbon cloth was placed in DCM solution with 10% PEI at 50 °C for 2h.

Supplementary Figures:

Fig. S14 Response currents of three devices with different counter electrodes at ~47 KPa pressure.

5. It is noted that the authors did not include several plots that contain crucial characterization information for a piezoionic generator. Specifically, the plots "Ion flux vs. Time," "Ionic conductivity vs. electrolyte concentration," and "Diffusion Coefficient vs. electrolyte

concentration" were absent from the manuscript.

Response:

We are very grateful for the reviewer's careful reviewing and kind comments. Fig. S8 shows the plot "Current vs. Time" which we think can represent the relationship between ion flux and time due to the correlation between net ion flow and electrical current. Fig. S15 shows the plots "Ionic conductivity vs. electrolyte concentration" and "Diffusion Coefficient vs. electrolyte concentration". As can be seen, the conductivity increases with the concentration but reaches saturation at around 4 M LiCl. Whereas the diffusion coefficient decreases with concentration in the concentration range of 1-6 M LiCl due to the increasing viscosity of the solution. In our system, we cannot experimentally distinguish the contribution of cations and anions. For reported piezoelectric generators, their performances are strongly dependent on the hydrogel itself. Therefore, the diffusion coefficient and ion conductivity are quite significant for the transportation of ions. But for our device, the performance is strongly relative to the electrodes. Many experiments (Fig. 2a, b, e, f and Fig. S7-S11) proved that the electrode strongly decides the output performance.

We have modified the manuscript to provide those plots (Fig. S8 and Fig. S15).

Revision:

Supplementary Figures:

Fig. S8 Generated current as a function of time during a standard compression process, which also represents the characteristics of instantaneous ion flux.

Fig. S15 Electrochemical characterization of hydrogel electrolytes. **a**, EIS curves of the PVA hydrogel infused with different concentrations of LiCl. **b**, Ion conductivities of PVA hydrogel infused with different concentrations of LiCl. **c**, Relationship between Z' and $\omega^{-1/2}$. **d**, Diffusion coefficients calculated from the EIS data. The measured ion conductivities and diffusion coefficients align well with reported data^{4,5,6,7}.

6. Regarding the PEI experiment, it may not be necessary to include it as a main figure since Figure 2a has already demonstrated the significance of ion selectivity. Moreover, the schematics in Figure 2e are identical, which should be a typo.

Response:

We are very grateful for the reviewer's careful reviewing and kind comments. According to this comment, we have modified the color of Fig. 2e for better illustration. As for the PEI experiment, we agree with the reviewer that it provided another data to support the significance of the electrode interface selectivity. Here, for the richness of the data and to demonstrate the versatile regulating strategies, we think it can be maintained in the main text.

Revision:

Main text:
Results

...e, Schematic illustration of the process of PEI treatment on ACC, altering the ion adsorption characteristics of ACC...

7. To ensure a comprehensive evaluation of the device's stability, it is recommended that the authors provide data characterizing the performance after subjecting it to long-term rinsing in water or exposure to air. Considering the potential dehydration of the hydrogel over time, which could lead to a decrease in performance, this information is crucial for understanding the device's long-term viability.

Response:

We are very grateful for the reviewer's careful reviewing and kind comments. Water soaking or evaporation is indeed a big challenge for a practical application, which is the issue for all hydrogel-based devices. In this work, we would agree that the PVA hydrogel is not a good anti-dehydration hydrogel. Long-term direct exposure to air can result in severe dehydration, despite using high-concentration lithium chloride to maintain its high moisture absorption. Therefore, we tested our device with the encapsulation by a waterproof membrane (Parafilm). The mass change and electrical change were recorded. The encapsulation can improve the stability. But future efforts are still highly required to solve this issue. Using glycerol as a co-solvent or implementing proper encapsulation would be a preferable approach. We are actively working on altering the operational form of the device to make it more amenable to better encapsulation.

We have added the result of the durability test as Fig. S24.

Revision:

Supplementary Figures:

Fig. S24 Dehydration of hydrogel energy converters in the atmosphere. One sample was encapsulated with a layer of Parafilm, while the other sample was exposed directly to the air at a temperature of 27°C and relative humidity of 59%. The mass change is represented by the blue curves, while the red curves depict the output change. It is evident that encapsulation can improve the stability. But future efforts are still required to fully solve this issue.

8. To optimize the device performance, it is crucial to consider the concentration of the LiCl solution, activation time of the ACC electrodes, and applied pressure. It is recommended that the authors explore whether there are specific threshold values for the concentration and activation time that yield optimal performance. Additionally, the authors should address the reason behind the saturation of current with increasing pressure.

Response:

We are very grateful for the reviewer's careful reviewing and kind comments. We have conducted additional experiments, as detailed in Fig. S16. Firstly, we increased the concentration of lithium chloride. Interestingly, the electrical output of the device decreased at 6M concentration. This can be attributed to the fact that at this concentration, the electrolyte's conductivity did not significantly improve, consistent with previous literature reports. However, at 6M, the PVA hydrogel exhibited much higher viscosity, as shown in the figure. It is possible that the functional groups on the ACC surface may be shielded, as observed with PAAM hydrogels (as mentioned in the first question from reviewer 1). Therefore, a 4M lithium chloride concentration appears to be an ideal value.

Fig. 2 The viscosity comparison of PVA hydrogels swollen in 4M (left) and 6M (right) LiCl solutions was conducted. It can be observed that the hydrogel swollen in 6M LiCl demonstrated a certain level of viscosity.

Next, we increased the activation time of the electrode. The drawbacks of increasing the activation time have been described in the literature (Nanoscale, 2016,8, 10406-10414). In our tests, samples with activation times of 30 minutes and 45 minutes exhibited nearly identical output currents, slightly higher than those activated for 15 minutes. This result can be expected, considering that the activation current density at constant oxidation voltage significantly decreases after approximately 20 minutes.

Finally, Fig. 3a also indicates that the output current reaches a saturation point as the pressure increases. This could be attributed to two reasons: (i) the ACC-hydrogel contact interface area

reaches saturation at high pressure, and (ii) the deformation of pyramid hydrogel reaches saturation and excessive pressure leads to overall deformation of the bottom hydrogel.

According to this comment. We have added Fig. S16 to show the measured data.

Revision:

Supplementary Figures:

Fig. S16 Current outputs from devices with various processing conditions. The device exhibits the highest output performance when the concentration of lithium chloride is 4M. Further increasing the concentration of lithium chloride causes the PVA hydrogel to become sticky, resulting in a decrease in electrical output. Extending the activation time to 30 minutes can enhance the output current. However, during the electrical activation of ACC, the activation current significantly decreases after approximately 17 minutes. Therefore, samples with activation times of 30 minutes and 45 minutes exhibit similar output currents.

9. In Figure S14, the application did not show the electrical connection to the mechanoionic generator. The authors should add the schematics for the application circuit and realistic photos of the generator connected to the application circuit.

Response:

We are very grateful for the reviewer's careful reviewing and kind comments. We have added them to Fig. S21.

Revision:

Supplementary Figures:

Fig. S21 **a**, Equivalent circuit of the power system (C: capacitor; PMC: power management chip, CJMCU 3108) **b**, Photos of the generator connected to the circuit.

10. Figure 4 currently lacks sufficient information. The authors are suggested to include additional data to enrich its content, such as the animal experiment results or other applications.

Response:

We are very grateful for the reviewer's careful reviewing and kind comments. We have revised the last paragraph of the Result section, Fig.3 and Fig. 4 to incorporate additional details (the animal experiment results).

Revision:

Main text:

Results

Fig. 3 | Output performance of the hydrogel generator... **f**, Comparison of this device with state-of-the-art SDC-TENG³⁸, EMG³⁹, TENG⁴⁰, and PENG⁴¹ in the domains of internal resistance, current density, softness, transferred charge, and density. **g**, Current outputs of this hydrogel generator under various frequencies of mechanical input, as compared with those of TENG and EMG. The test parameters are listed in Supplementary Table S3...

... Fig. 4b presents an exploded view schematic image and a photograph of SAP, which includes an adhesive bandage substrate, Ag/AgCl reference electrode, ACC electrode, arched PVA hydrogel electrolyte, CC electrode, drug-loaded cefazolin modified carbon cloth (CC-CZ) patch, and wet cotton immersed in PBS for drug diffusion and protection. The SAP was employed to promote wound healing in infected mice. After anesthesia and cleaning the operative area, we generated two symmetrical full-thickness skin incisions in the dorsal flank region of mouse and then infected with *Staphylococcus aureus* (*S. aureus*). Subsequently, the SAP was fixed to cover the wound on the right side, while a commercial adhesive bandage was fixed to cover the other wound as control (Fig. S31). Movements of the mouse induce stretching of the bandages, which deform the arched hydrogel and change the physical contact between hydrogel layer and the working electrode. The generated current flows from ACC through the silver/silver chloride, the PBS-containing cotton cloth, to the drug-loaded CC-CZ and back to CC, as described in Fig. S32. The sizes of the wounds were recorded every two days to evaluate the treatment efficiency, as presented in Fig. 4c and 4d. Evidently, the area of the wound covered by SAP is smaller than the area of the other wound covered by commercial bandage

during the nine-day healing period. On the last day, the wound area covered by the SAP and regular patch was reduced to 13% and 45% of the initial wound area, respectively. The significant improvement of the wound healing by SAP indicates effective drug release driven by mechano-electrical energy conversion. Indeed, the high current output, low voltage, and low impedance of the hydrogel generator could enable diverse applications in controlled drug delivery as the chemistry and dosage can be further engineered.

Fig. 4 | Application in a drug-releasing device. **a**, In vitro demonstration of electrically controlled drug release with "on-off" cycles driven by the hydrogel generator. The "on" state (blue) represents tapping the device to provide output current. **b**, Schematic (upper) and photography (bottom) of the self-powered drug-releasing patch (see details and results of the animal experiment in Methods and Fig. S29-S30). **c**, Photographs of the wound during the healing process with a standard patch (upper) and the SAP (lower). Scale bar: 5 mm. **d**, The relative area of wounds as a function of healing time. On the last day, the wound area covered by the normal patch and SAP was reduced to 32% and 11% of the initial wound area, respectively.

11. In figure S22, the number of calibration data points needs to be increased. The author should provide more data points uniformly in all ranges.

Response:

We are very grateful for the reviewer's careful reviewing and kind comments. We have increased the number of calibration data points and modified Fig. S30.

Revision:

Supplementary Figures:

Fig. S30 Standard calibration curve of cefazolin sodium.

12. Instead of solely comparing its performance with other technologies, the authors should focus on discussing the feasibility and practicality of this approach in the application for drug delivery.

Response:

We are very grateful for the reviewer's careful reviewing and kind comments. We have revised the last paragraph of the Result section to include a discussion on the usability and applicability of this approach in drug delivery.

Revision:

Main text:

Results

Indeed, the biocompatibility, high current output, low voltage, and low impedance of the hydrogel generator could enable diverse applications in controlled drug delivery as the chemistry and dosage can be further engineered.^{1, 4.}

Reviewer #3 (Remarks to the Author):

(1) I wonder the generality of their system. how crucial is PVA hydrogel? would it work for other hydrogel also?

Response:

We are very grateful for the reviewer's carefully reviewing and kind comments. We have tried other hydrogels, as shown in Fig. S12 and discussion in the first question of reviewer 1. When using GelMa as the hydrogel, the system can obtain a higher output current under the same pressure.

Revision:

Results

...When the PVA hydrogel is replaced with a gelatin methacryloyl (GelMa) hydrogel, this phenomenon can still be observed, demonstrating the generality of the proposed mechanism (Fig. S12). When using PAAM for the hydrogel layer, however, the output current is significantly lower. In fact, PAAM hydrogel exhibit much greater viscosity than PVA and GelMA hydrogels. The distinct interfacial interactions may affect the surface functionality of ACC, which is worth future investigations³⁴.

Supplementary Figures:

Fig. S12 Current outputs of different hydrogels. Three different hydrogels were tested, i.e. PVA, PAAM, and GelMa hydrogels. All the hydrogels were soaked in 1M LiCl solution and air-dried before testing. The strain rate (16 mm/s) and interval time (10.8 s) were the same for all tests. Both PVA and GelMa hydrogels achieved milliampere-level output currents; PAAM hydrogel only showed microampere-scale currents. In fact, GelMa and PVA hydrogels do not exhibit high adhesion to ACC; in contrast, PAAM hydrogel, when taken out from the salt solution and air-dried for a period, exhibited high adhesion to ACC. The right figure (b) shows the mass change of the ACC electrode before and after contacting the PAAM and PVA hydrogel, respectively. After being pressed against the PAAM hydrogel, the ACC surface is coated with a layer of hydrogel or salt solution, which may affect the surface functionality. The larger output of GelMa hydrogel is probably because it is softer and hence a larger ACC-hydrogel interface

area at compressed state than that of the PVA hydrogel. Therefore, the PVA hydrogel was finally selected based on its following merits: (1) PVA hydrogel can be easily obtained with highly consistent mechanical properties through simple freeze-thaw cycles. This allows for better control of variables during the experimental process. (2) by increasing the solid content and the number of freeze-thaw cycles, PVA hydrogel with higher mechanical properties can be obtained. This is beneficial for the demolding process of the hydrogel and helps maintain its resilience, morphology, structure, and stable mechanical properties during subsequent compression-separation testing, enabling it to withstand higher pressures. (3) PVA hydrogel obtained through freeze-thaw cycles can be conveniently assembled, facilitating seamless integration with electrode carbon cloth. (4) the relatively simple and stable chemical structure of PVA is advantageous for analyzing the device performance.

(2) Figure 3e demonstrated the high durability of their device. nevertheless, I expect PVA hydrogel would be sticky and may adhere to the ACC electrode surfaces with repeating loads or high pressure. Also, the ACC surface may be pretty rough. Would pyramid tips stuck into the electrode surfaces after repeated pressing? some explanation on the soft/hard interface may be helpful by referring to the related literature (<https://doi.org/10.1002/adma.201902278>).

Response:

We are very grateful for the reviewer's carefully reviewing and kind comments. We did indeed observe this issue with PAAM hydrogels, as described in the first question from reviewer 1. Due to the viscosity of these hydrogels, a layer of hydrogel or salt solution adheres to the ACC surface after contact, which may lead to different interfacial chemistry. However, the PVA hydrogel swollen in 4M lithium chloride did not exhibit significant tackiness, as described in the eighth question from reviewer 2. We did not observe significant attachment of PVA hydrogel or water film to the ACC. The literature suggested by the reviewer was highly beneficial for understanding this issue, and we have cited it as Reference 34 in the main text.

Revision:

Main text:

Results

When using PAAM for the hydrogel layer, however, the output current is significantly lower. In fact, PAAM hydrogel exhibit much greater viscosity than PVA and GelMA hydrogels. The distinct interfacial interactions may affect the surface functionality of ACC, which is worth future investigations³⁴

(3) what is the estimated mechanical-electricity conversion efficiency?

Response:

We are very grateful for the reviewer's careful reviewing and kind comments. According to our calculation, the estimated mechanical-to-electrical energy conversion efficiency is 5%. The

detailed calculation method has been provided in the Supplementary Note section.

Revision:

Supplementary Note 2:

Estimated mechanical-electricity conversion efficiency: We calculated the mechanical-electricity conversion efficiency according to the following equations.

$$E_M = \int F dx \quad \text{S15}$$

Here E_M means the input mechanical energy; F means the force during the compression process; x means the displacement of the ACC electrode.

$$E_E = R \int I^2 dt \quad \text{S16}$$

Here E_E means the output electrical energy; R is the internal resistance (122 Ω); I is the current and t is the time.

$$\eta = \frac{E_M}{E_E} \times 100\% \quad \text{E17}$$

Based on the force-displacement curve and current-time curve of the CC-PVA 4M LiCl-ACC device under an applied pressure of 85 kPa, we have calculated that the energy conversion efficiency of the device is approximately 5%.

REVIEWER COMMENTS

Reviewer #1 (Remarks to the Author):

The authors have carefully revised the manuscript and most of the issues that I concerned have been addressed. Several minor issues should be addressed.

- 1.The preparation method of the PAAM hydrogel and the pyramid structure should be provided. What is the interaction between PAAM gel and ACC? How did PAAM gel adhere on ACC?
- 2.Did the mechanical strength influence the performance of generator? The mechanical properties of three different hydrogels should be investigated and discussed.
- 3.As shown in Fig. S12, why did the gels require an air-dried process? This process of different kinds of gels may not ensure uniformity.

Reviewer #2 (Remarks to the Author):

The authors have well addressed most of the proposed comments, but the following responses need further clarifications.

1. For the modified figure S15a, the EIS data seems incomplete. Please either include the complete EIS data or adjust the x-axis limit to fully showcase the data.
2. Modified Figure 3e has zoom-in views of different portions of the waveform. Please mark these portions on the complete waveform.

Reviewer #3 (Remarks to the Author):

The authors have well addressed my comments. It is now publishable.

Dear Reviewers,

Thank you very much for your consideration of this work and the insightful comments. The comments and suggestions are helpful to improve the paper quality. According to the insightful suggestions, the manuscript has been revised carefully, and the changes are marked in **yellow**. The point-to-point response to each reviewer is given below.

Reviewer #1

1. The preparation method of the PAAM hydrogel and the pyramid structure should be provided. What is the interaction between PAAM gel and ACC? How did PAAM gel adhere on ACC?

Response:

We sincerely appreciate your kind comments. The preparation method of the PAAM hydrogel is shown here. 33 wt% AAM solution was mixed with 0.1 wt% MBAA and 0.5 wt% photoinitiator 2529. The precursor solution was poured into a mold and cured under UV light. Regarding the interfacial interactions, hydrogen bonding between oxygen-containing functional groups on ACC and amide groups on PAAM hydrogel and the van der Waals interactions involving loose polymer chains may contribute to the adhesion of PAAM hydrogel on ACC. However, exact physicochemical interactions and their contributions to the electrical responses for PAAM-based devices (or GelMA-based devices) would require another systematic study and are subject to separate reports.

Revision:

Fig. S12:

PAAM: 33 wt% AAM solution was mixed with 0.1 wt% MBAA and 0.5 wt% photo-initiator 2529. The precursor solution was poured into a mold and cured under UV light.

2. Did the mechanical strength influence the performance of generator? The mechanical properties of three different hydrogels should be investigated and discussed.

Response:

We sincerely appreciate your kind comments. We agree that the mechanical properties of hydrogels may influence the mechanoionic responses of the device. We added a preliminary FEM simulation on the effect of modulus of the hydrogel on the strain gradient. We can see that a lower compression modulus leads to a higher strain gradient under the same external force, which may result in a higher output current. However, other chemical and physical factors may couple with the mechanical properties and contribute the complex electrical responses. A more systematic investigation would be helpful but is beyond the scope of the current study and is subject to a separate report.

Revision:

Supplementary Figures:

Fig. S2 e-f), FEA results of compressive strain in hydrogels with different compressive moduli (60 kPa for e and 100 kPa for f) under the same external force (~28 mN). A lower compression modulus may lead to a higher strain gradient, which would generally result in a higher output current. However, other chemical and physical factors may couple with the mechanical properties and contribute the complex electrical responses.

3. As shown in Fig. S12, why did the gels require an air-dried process? This process of different kinds of gels may not ensure uniformity.

Response:

We sincerely appreciate your kind comments. And we are sorry about causing the confusion. To keep the surface of the hydrogel dry after removing it from the electrolyte solution, we use lint-free paper to absorb the water from the surface. Once the sample is placed on the testing stage, we need to set up the testing system, which leads to a brief period of air drying. Indeed, the performance of the hydrogel devices is stable over a period of time in the atmosphere (Fig. 3), so we argue that dehydration is not a major issue for the present demonstration. However, further encapsulation design would be helpful and subject to a separate work. We have modified the description in the text to avoid confusion.

Revision:

Fig. S12 (we changed the word “air-dried” to “wiped”)

Reviewer #2:

1. For the modified figure S15a, the EIS data seems incomplete. Please either include the complete EIS data or adjust the x-axis limit to fully showcase the data.

Response:

We are very grateful for the reviewer’s kind comments. The updated figure was placed in Fig. S15. We adjusted the x-axis limit.

Revision:

Supplementary Figures S15:

2. Modified Figure 3e has zoom-in views of different portions of the waveform. Please mark these portions on the complete waveform.

Response:

We are very grateful for the reviewer's kind comments. We have marked these portions on the complete waveform.

Revision:

Main text: Fig. 3

Reviewer #3:

The authors have well addressed my comments. It is now publishable.

Response:

We are very grateful for the reviewer's kind comments.

REVIEWERS' COMMENTS

Reviewer #1 (Remarks to the Author):

The authors have properly revised the manuscript

Reviewer #2 (Remarks to the Author):

The authors have addressed our comments well